# Automatically Extracting Rubber Tree Stem Shape from Point Cloud Data Acquisition Using a B-Spline Fitting Program

Tuyu Li , Yong Zheng *, Chang Huang, Jianhua Cao, Lingling Wang and Guihua Wang

Rubber Research Institute, Chinese Academy of Tropical Agricultural Sciences, Haikou 571101, China; lituyu@catas.cn (T.L.)
* Correspondence: zhengyong07@163.com; Tel.: +86-0898-66961272

**Abstract:** Natural rubber is an important and strategic raw material, used in tires, gloves, and insulating products, that is mainly obtained by cutting the bark of rubber trees. However, the complex contour curve of the rubber tree trunk is hard to fit using a tapping machine. Thus, a trunk contour curve collection would be useful for the development of tapping machines. In this study, an acquisition system based on laser-ranging technology was proposed to collect the point cloud data of rubber tree trunks, and a B-spline fitting program was compiled in Matrix Laboratory (MATLAB) to extract the trunks' contour curves. The acquisition system is composed of power, a controller, a driver, a laser range finder, and data transmission modules. An automatic extraction experiment on the contour curves of rubber tree trunks was carried out to verify the feasibility and accuracy of using the acquisition system. The results showed that the degree of rubber tree trunk characteristic recognition reached 94.67%, which means that the successful extraction of the rubber tree trunk contour curves and the B-spline fitting program are suitable for the extraction of irregular curves of rubber tree trunks. The coefficient of variation of repeated collection was 0.04%, which indicates that changes in relative positions and acquisition directions have little influence on the extraction and the accuracy of the acquisition system, which are high and stable. Therefore, it was unnecessary to adjust the position of the acquisition device before the collecting process, which helped to improve the efficiency of acquisition considerably. The acquisition system proposed in this study is meaningful to the practical production and application of agroforestry and can not only improve the precision of the rubber tapping process by combining with an automatic rubber tapping machine but can also provide technical support for the prediction of rubber wood volume and the development of ring-cutting equipment for other fruit trees.

**Keywords:** rubber tree; laser-ranging; point cloud data; contour curve; B-spline; MATLAB

## 1. Introduction

Natural rubber is a significant material in national defense and industry and is widely used in the fields of aerospace, medicine, and healthcare, with more than 7000 kinds of downstream products [1,2]. Affected by temperature and climate, natural rubber is usually obtained by tapping rubber tree bark at 4–7 a.m. [3]. The operation of tapping is a process with high precision, strength, and technical requirements, as the tapping depth must be controlled in the range of 1.2–1.8 mm with a tapped bark thickness <1.4 mm [4]. The key to guaranteeing quality and production is to precisely control the tapping depth by fitting the tap to the contour of the rubber tree. The main reason is that a deeper tapping depth will cause dead bark to become diseased and decrease the economic life of the rubber tree, while a shallower depth cannot destroy enough laticiferous tubing and reduces the production of natural rubber [5–7]. However, the coarse contours of rubber trees are too complicated to fit using existing equipment, which mainly relies on mechanical deformation and leads to low precision and undesirable effects during the tapping operation [8–10]. Therefore,

technologies must be developed for the extraction of rubber tree contours to improve the accuracy of tapping.

Critical to the process of extracting a rubber tree's contour curve is sufficient point cloud data. The acquisition methods for point cloud data include laser-range scanning, ultrasonic detection [11,12], and visual identification [13,14]. Laser-range scanning is the main way to collect point cloud data in the forestry field and is used to analyze the distribution and overall contours of plants.

The distribution of forestation is important to the forest inventory. Hyyppä et al. [15–17] have conducted research on various laser scanning techniques, such as with a backpack, hand-held, and with a UAV (unmanned aerial vehicle), and compared these techniques in boreal forest conditions. The results indicated that backpack and handheld laser scanning techniques are more suitable for estimating the stem volumes of individual trees, and above-canopy UAV laser scanning techniques are not yet sufficient for stem attribute detection. Meanwhile, the handheld laser scanning system provides more possibilities for forest inventory, and many relevant tree attributes, such as tree detection, stem position, and tree height, can be collected with the handheld technique [18]. However, the performance of UAV laser scanning offers new opportunities to estimate forest stock volume. Puliti et al. [19] found a method to measure tree attributes without the use of field data, and the results showed that estimates based on UAV laser scanning data were within the 95% confidence interval.

The overall contours are mainly used to estimate the wood volume and the growth situation of a forest. Holmgren et al. [20] estimated stem diameters using a mobile laser scanning system; the results showed that accuracy was high for the basal area–weighted mean stem diameter and the basal area and the relative RMSEs were 3.4% and 8.5%, respectively. Lin et al. [21] identified stems, branches, and leaves in a forest model combined with three kinds of K-nearest neighbor algorithms. The results showed that the ratios of eigenvalues for the point cloud plane, perpendicularity, and curvature played an important role in point cloud classification. Pires et al. [22] also used the same method to extract the diameter at breast height, the contour curve, and the volume of the stem; however, the scanning results were easily disturbed by branches and leaves. Those results also found that recognition accuracy is diverse at different acquisition distances. Chianucci et al. [23] used terrestrial laser scanning to determine the volumes of stems and crowns of aspen, and the results proved that laser scanning is an effective means of measuring forest information in a nondestructive manner. Saha et al. [24] installed a laser ranging device on a tractor and collected the point cloud data of fruit trees to extract the parameters of height and width; the RMSDs were 2.09% and 0.91% for height and width, respectively. Those researchers observed that the data became biased due to the perturbating influences of the range and incidence angle of the laser beam. Stal et al. [25] extracted the diameter at breast height and evaluated the volume of each tree using Computree 3.0 software, and the results were three times shorter than manual measurement and five times shorter than conventional static terrestrial laser scanning techniques.

Although some researchers have used ultrasonic technology to extract the contours of tree trunks, there are many pits and cracks on the surface of rubber tree trunks, especially in the tapping panel, that will affect the reflection quality of an ultrasonic wave. As for the method of visual identification, it is too expensive and unreliable; thus, the method was not considered in this study.

After the collection of enough contour point cloud data, the data need to be processed to acquire the correct contour curves. There are several methods of irregular curve fitting, such as the polynomial equation [26], Bezier spline [27,28], and B-spline [29]. Among these, B-spline is the most suitable for complex curve fitting and can be modeled in Python, MATLAB, and AutoCAD [30,31]. Based on the method of B-spline modeling, Ma et al. [32] established a 3D model of crowns and stems to analyze the growth tendencies of fir trees in all possible directions; the results coincided with actual measurements and proved useful for plant growth modeling. Wu et al. [33] developed an editable branch, leaf, and root model that can be used to simulate the growth of plants, and the data could be conveniently

uploaded to the Internet after compression. Li et al. [34] realized the parametric modeling of bowl-shaped, peduncle-shaped, and lumpy tree knots, and their results had a certain significance for evaluating wood quality and improving the realism of the surface textures of knot models. Yang et al. [35] set up a root model of rice combined with an L-system to study more about the form, structure, and function of rice. A new integration method was used to obtain the angles between roots to realize a visual simulation of rice root systems. In addition, the B-spline model has been applied to the machining and medical fields. Zhang et al. [36] proposed a new shape-modeling method called Hierarchical D-NURBS and developed an editable engraving interface by controlling points, lines, curvatures, and constraint features. Furthermore, Chen et al. [37] modeled the human heart to analyze its function and identify cardiovascular disease according to the heart's feature points. Thus, the B-spline curve fitting method was adopted to extract rubber tree trunks' contour curves in this study.

Research on the contour curve acquisition of rubber trees remains limited at present, and there are only references to other technologies of contour extraction. Nevertheless, the accuracy of automatic tapping machines is in urgent need of improvement, so contour curve acquisition technologies for rubber trees need to be developed. In this research, we propose an automatic extracting method based on laser ranging technology and conduct an experiment to test the feasibility of the method and support the research of automatic rubber-tapping equipment.

## 2. Materials and Methods

### 2.1. Rubber Tree Trunk Characteristics

Rubber trees are suitable for planting within 15° N, but in China, the majority of rubber trees are located in Yunnan, Hainan, and Guangdong, where the latitude reaches 18–24° N. The rubber trees frequently have a rough surface and an irregular stem contour as a result of the effects of typhoons, rainfall, light, soil nutrients, and other elements, which increase the difficulty of mechanical tapping.

The rubber tree trunk utilized in the experiment was acquired from Hainan clone Reyan 7-33-97, which has been the dominant cultivar of rubber tree for numerous years. The trunk has a radius of around 93 mm. This trunk's surface kept its original tapping state. As shown in Figure 1, the following characteristics were clearly visible on the surface of the trunk: tapping panel (the area of tapping that served as a standard for subsequent tapping), crack, hollow, front tapping line (start point of tapping, which was processed by rubber tappers before tapping), and back tapping line (finish point of tapping, which was processed by rubber tappers before tapping and used for latex drainage). Because these characteristics were critical tapping parameters that the tapping machine needed to detect exactly before the tapping procedure, the recognition rate of these features was used as one of the assessment indices in this study.

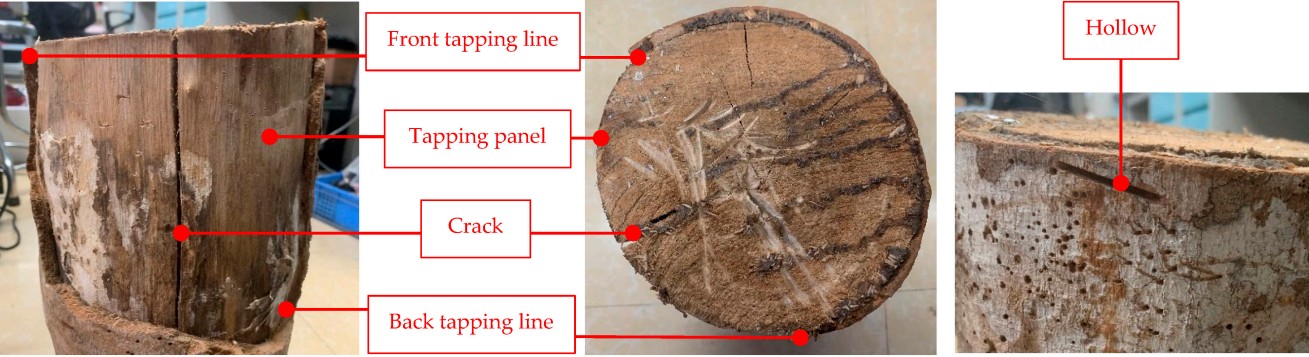

**Figure 1.** Characteristics of rubber tree trunk.

### 2.2. B-Spline Curve Fitting Processing in MATLAB

The contour curve of a rubber tree is difficult to characterize using a regular pattern due to its complicated contour forms. A random section of a rubber tree is considered as an irregular curve with finite discrete points in this study. The contour curves of the rubber tree are recovered using coordinate transformation and extracted by the B-spline curve fitting program in MATLAB [38]. Figure 2 depicts the acquisition of the point cloud data. The laser ranging finder moves at a steady speed along the circular track to collect contour point cloud data from the rubber tree.

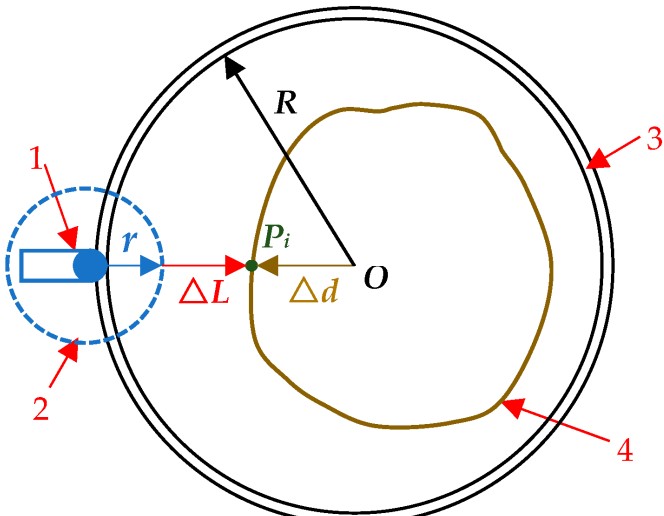

**Figure 2.** Method of point cloud data acquisition: 1—laser range finder; 2—zero radius of the finder; 3—acquisition trajectory; 4—rubber tree.

The distance between each contour point $P_i$ on the rubber tree contour and the center point $O$ may be computed using Equation (1), which varies with the distance between the laser finder and the rubber tree trunk.

$$\Delta d = R - r - \Delta L, \tag{1}$$

where

$\Delta d$—distance between the contour point and center point $O$, expressed in mm;
$R$—scanning radius related to the length of scanning arm, expressed in mm;
$\Delta L$—point cloud data collected, expressed in mm;
$r$—zero radius of laser ranging finder, expressed in mm, where $r$ = 200 mm.

The motor's starting/stopping position is delayed, and some of the point cloud data may have been delivered to the computer prior to starting/stopping. Therefore, before importing point cloud data into the program, duplicate data at the beginning and end of the data will be eliminated.

The angular velocity and data transmission speed of the laser range finder are set to a constant value so that the contour point can be thought of as a set of poles distributed uniformly around the rubber tree in 360°, with the center point, polar radius, and increments in the polar angle being $O'$, $\Delta d$, and $\Delta\theta$, respectively, as shown in Figure 3.

The polar angle increment is equal between adjacent poles; thus, an arithmetic sequence is introduced, with the first term being 0, the last term being $2\pi$, and the tolerance being $\Delta\theta$. The increment in polar angle (tolerance) is calculated using Equation (2) if the amount of point cloud data was $n + 1$, expressed as $P_i$ ($i = 0, 1, \ldots, n$):

$$\Delta\theta = 2\pi/(n + 1), \tag{2}$$

where

$\Delta\theta$—increment in the polar angle, rad.

The contour point $P_i$ can be described in polar coordinates by Equation (3), which can then be translated into rectangular coordinate points in MATLAB, as follows:

$$P_i = (\rho_i, \theta_i) = (\Delta d_i, i \cdot \Delta\theta),\tag{3}$$

where

$\rho_i$—polar radius of contour point, mm;
$\theta_i$—polar angle of contour point, rad.

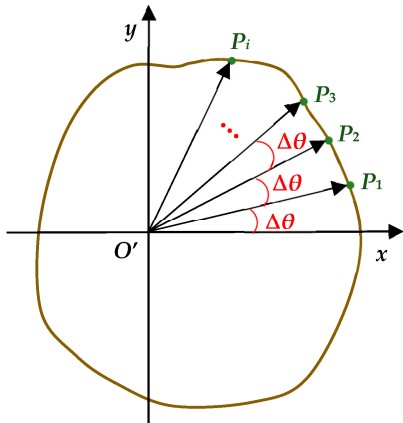

**Figure 3.** Point cloud data represented by polar coordinates.

Following the acquisition of the contour point coordinates, B-spline curve fitting will be performed to extract the contour curve of the rubber tree. The contour point $P_i$ is used as the interpolating point of the spline curve, and the control points of curve can be generated using the Thomas algorithm [39]. Equation (4) represents the third-degree B-spline and Equation (5) represents the equation of each B-spline segment [23] (pp. 234–299):

$$Q_i(u) = \frac{1}{6}(u^3\ u^2\ u^1\ 1)\begin{bmatrix} -1 & 3 & -3 & 1 \\ 3 & -6 & 3 & 0 \\ -3 & 0 & 3 & 0 \\ 1 & 4 & 1 & 0 \end{bmatrix}\begin{bmatrix} P_1 \\ P_2 \\ P_3 \\ P_4 \end{bmatrix},\tag{4}$$

$$\Rightarrow Q_i(u) = [P_0(1-u)^3 + P_1(3u^3 - 6u^2 + 4) + P_2(-3u^3 + 3u^2 + 3u + 1) + P_3 u^3)]/6,\tag{5}$$

where

$Q_i$—the expression of curve segment, $i = (1, 2, \ldots, n)$;
$u$—local parameter, $u \in [0,1]$.

Therefore, the coordinates of fitting points $FP_i(x, y)$ can be calculated using Equation (6):

$$FP_i(x, y) = [P_i(1-u)^3 + P_{i+1}(3u^3 - 6u^2 + 4) + P_{i+2}(-3u^3 + 3u^2 + 3u + 1) + P_{i+3}u^3)]/6\tag{6}$$

An automatic B-spline fitting algorithm was established based on Equations (1)–(6) to extract the contour curve of the rubber tree trunk, as shown in Appendix A. The rationale of the algorithm is as follows:

- Read point cloud data and convert them to rectangle coordinates (Lines 2–9); "record3" is the name of the database used to record point cloud data, which should be modified before executing the program.
- All of the cloud points were used as knots, and the coordinates of the control points were calculated using the Thomas algorithm (Lines 11–20), which was written as a function named Chase_method.

- Establish the basic equations of B-spline (Lines 22–28).
- Calculate the coordinates of fitting points based on the basic equations (Lines 30–31).
- Since the contour curve of the rubber tree trunk is a closed curve, the initial and last points are the same, so in this algorithm, an end point is appended to the contour curve, which has no influence on the fitting result (Lines 35–37).

### 2.3. Acquisition System Adopted

An automatic acquisition system used for collecting the contour point cloud data of the rubber tree trunk was constructed using the extraction method. The system comprises five major modules (shown in Figure 4):

- As a power module, the MW-230-115 switching power supply delivered consistent 24V DC voltage to devices in the acquisition system.
- The control module incorporates a Haiwell Company ACM120R Series PLC programmable controller and a speed regulator for determining the acquisition direction and speed.
- The executive module includes an EzM-42XL Series servo motor and servo drivers that are used to drive the finder around the rubber tree; the output torque of the servo motor is 0.65 N·m.
- The acquisition module was a HG-C1200 Series laser ranging finder produced by Panasonic, with an accuracy of 0.07 mm, a center radius ($r$) of 200 mm, and an effective range of 120–280 mm (the distance between the rubber tree and finder cannot exceed this range).
- A Bluetooth transmission device with an 800 Bd baud rate was used to transmit the point cloud data, and the data signal was transformed into a keyboard signal capable of automatic recording.

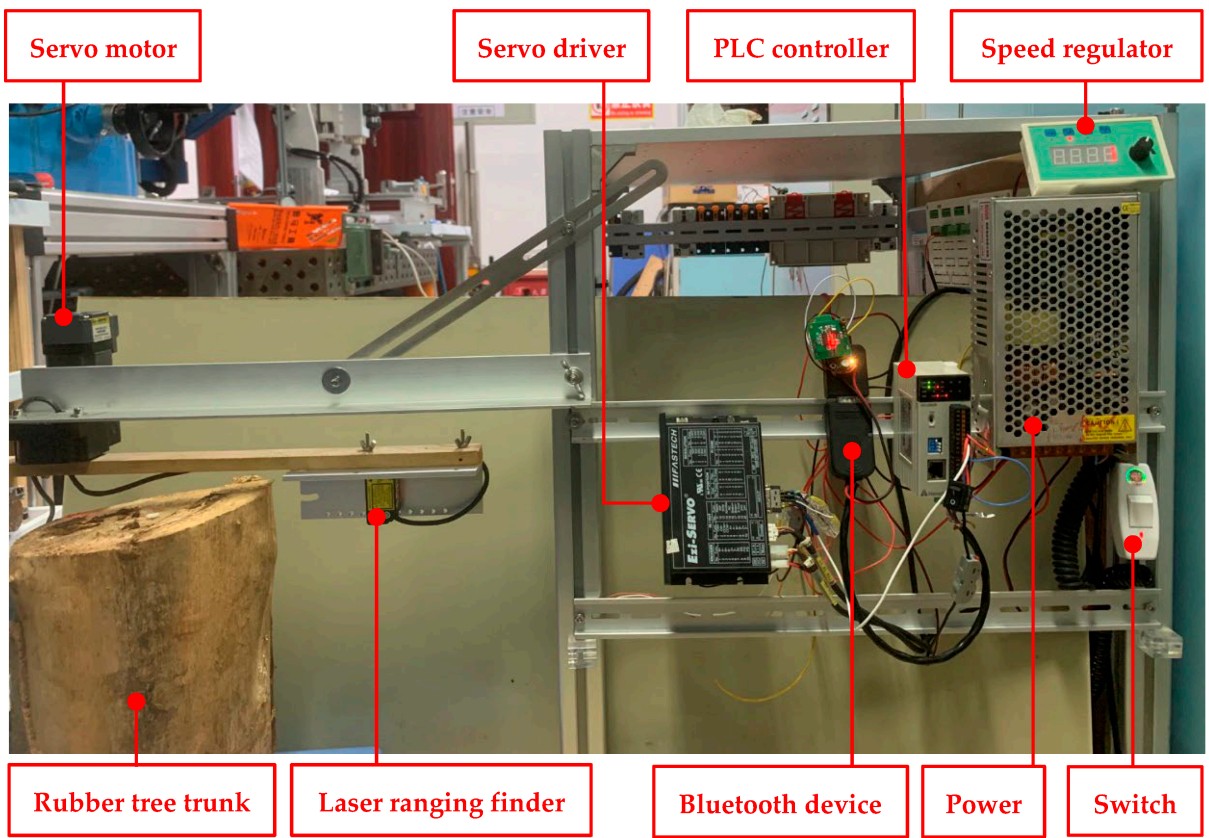

**Figure 4.** Structure of the acquisition system.

As shown in Figure 5, the operation of this system can be divided into three phases: power flow, control flow, and data flow. To avoid repetitive collection and errors in the contour point cloud data, the acquisition and data transmission on–off signals must be controlled precisely. The established control program can realize the following functions:

- Drive the servo motor at a constant velocity to guarantee that the point cloud data are captured uniformly;
- Continuously monitor the servo motor's angle and break down the operation when the angle reaches 360° to avoid the appearance of repeated collection points;
- Transmit contour point cloud data to the PC at a set rate. The operating principle of the acquisition system is shown in Figure 5. The program of PLC controller was compiled in the visual programming environment of Haiwell-Happy V2.2.9 and it met the requirements after verification.

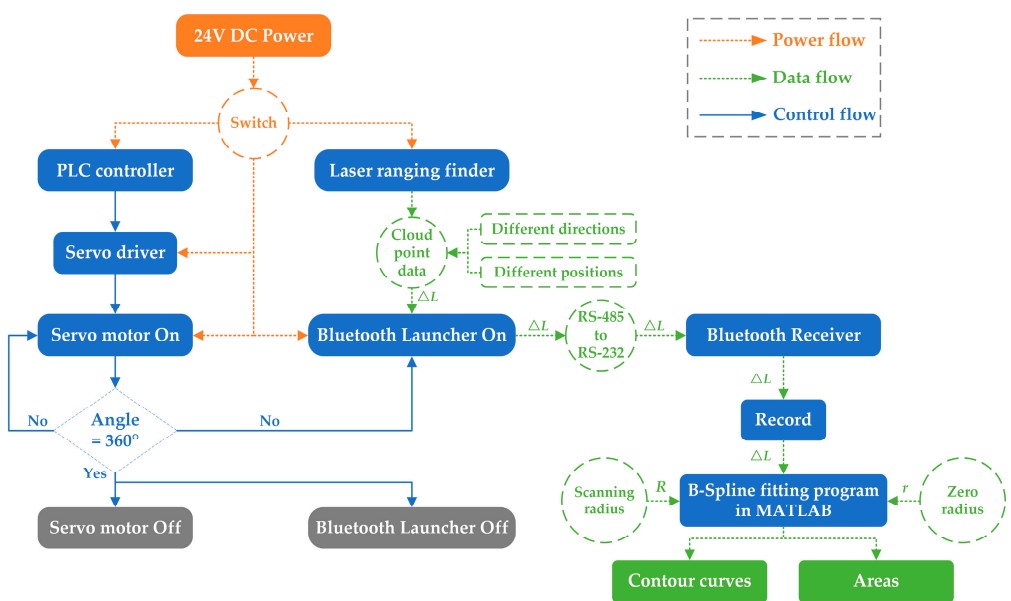

**Figure 5.** Operational process of the acquisition system.

A repeating experiment was conducted based on the operational process of the acquisition system to verify the feasibility and accuracy of this extraction method. The laser ranging finder was driven around the rubber tree trunk to collect contour points in different directions, and each direction had five sets of contour points. Following that, the position of the rubber tree was altered twice, and five sets of contour points were extracted in each position in different directions. As demonstrated in Figure 6, the positions indicated refer to the relative position of the trunk and center point.

### 2.4. Method of Analysis

The accuracy and efficacy of the extraction method was evaluated in this study using three indicators: the degree of characteristics recognition (*DCR*), the variable coefficient ($\sigma$), and the confidence coefficients of direction and position (*p*). The characteristics of the rubber tree trunk could be expressed as follows: ① tapping panel; ② front tapping line; ③ back tapping line; ④ crack; and ⑤ hollow. These characteristics were used for comparison with the acquired contour curve and to calculate the degree of characteristics recognition via Equation (7):

$$DCR = (n_1/n) \times 100\%, \tag{7}$$

where

$n_1$—number of features recognized;
$n$—number of features.

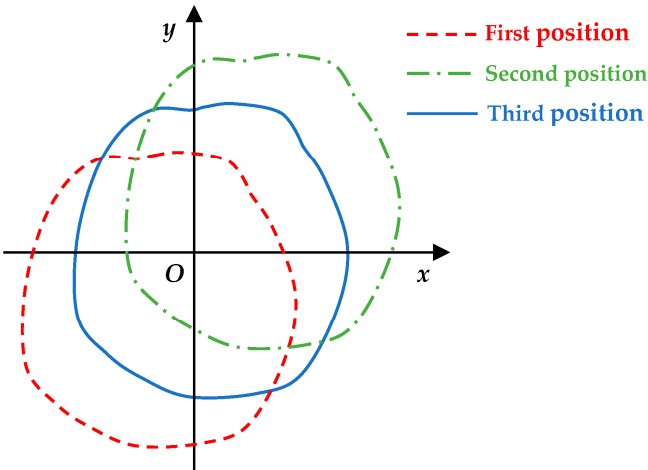

**Figure 6.** Relative position of the rubber tree: *O*—center point.

Following the extraction of the contour curve of the rubber tree trunk, the area of the contour curves was calculated using the area computation program for closed curves in MATLAB. The mean area ($\mu$) of the contour curve was calculated using Equation (8) based on the statistical analysis method:

$$\mu = \frac{1}{N} \sum_{i=0}^{N} q_i, \tag{8}$$

where

*N*—the quantity of the acquired contour curve;
$Q_i$—the area of the contour curve, expressed in mm$^2$.

Based on Equation (7), the standard deviation can be calculated using Equation (9):

$$s = \sqrt{\frac{1}{N} \sum_{i=0}^{N} (q_i - \mu)}, \tag{9}$$

Subsequently, Equation (10) was used to acquire the variable coefficient, which was then used to evaluate the accuracy of the contour curve throughout the repeated experiments. The repeated accuracy of the contour curve was high if $\sigma \leq 0.05$, and it was low if $\sigma > 0.05$:

$$\sigma = s/\mu. \tag{10}$$

Statistical Product and Service Solutions (SPSS) 25 software was used to conduct two-factor analysis of variance. This software is designed for statistical analysis, data mining, predictive analysis, and decision support. Different directions (clockwise and counterclockwise) and positions (the first position, the second position, and the third position) were set as fixed factors in the experimental scheme designed in this study, and the area of the trunk's contour curve was set as the dependent variable at first. The analytical model was then divided into three parts: direction, position, and direction × position. Finally, after adjusting the analysis setting, the results of the two-factor analysis of variance were presented in SPSS.

## 3. Results

### 3.1. Results of Different Acquisition Directions

The point cloud data collected from different acquisition directions were transmitted into MATLAB software, and the extracted curves of the rubber tree trunk were fitted using the self-edited B-spline program. Figure 7 depicts the extracted curves.

According to the extracted contour curves, the characteristics were clear and the shape of the contour curves was similar to the rubber tree trunk used in the experiment, indicating that the extraction method and acquisition system proposed in this study are feasible and the acquisition accuracy meet expectations.

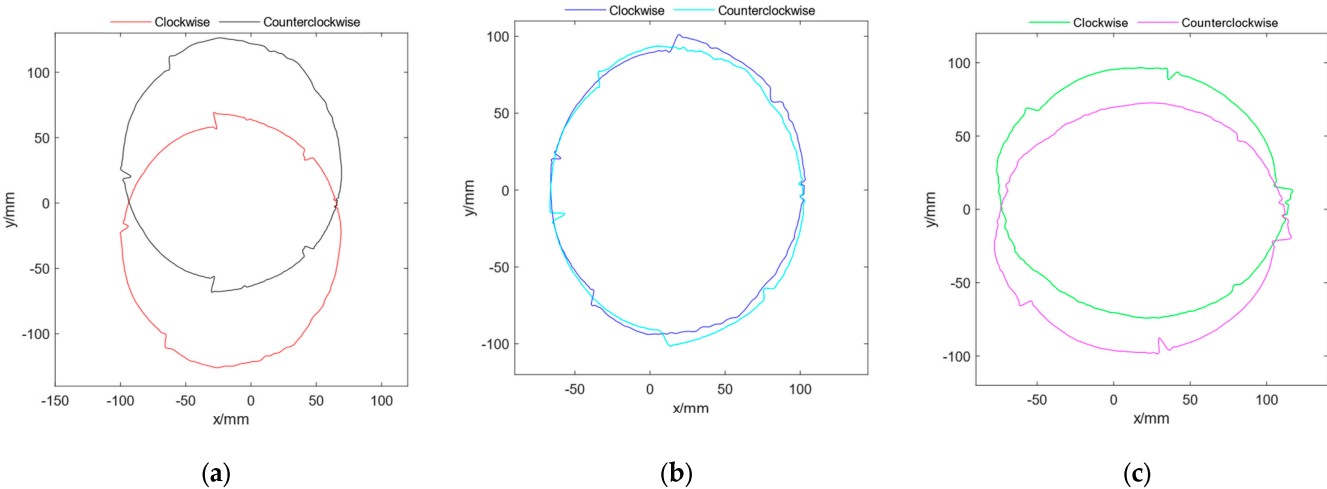

**Figure 7.** Results of contour curves in different acquisition directions: (**a**) the first position; (**b**) the second position; (**c**) the third position.

The successful extraction of the contour curve of the rubber tree trunk indicated that the MATLAB-compiled B-spline fitting program is acceptable for the extraction of irregular curves such as the contour curve of the rubber tree trunk. Furthermore, the fitting program is simple to use, only requiring the input of the contour's point cloud data, scanning radius, and zero radius of the laser ranging finder.

To differentiate the contour curves acquired from different directions in the same relative position, they were matched using different colors. It is also clear that the contour curves obtained from different directions in the same position are essentially consistent and that the two contour curves are axisymmetric relative to the x-axis. In addition, the center of the contour curves in opposite acquisition directions will be offset, but the degree of this offset is minimal at the second position, which may be connected to the shortest distance between the center of the trunk and the acquisition track. Although the contour curves obtained in different acquisition directions are offset, it appears that this has no influence on the contour curve results.

### 3.2. Results of Different Relative Positions

Following the validation of the acquisition system and the B-spline fitting program, more contour curves of the rubber tree trunk were acquired to further validate the repeating accuracy of the acquisition system, as shown in Figure 8. After repeated measurements, the results showed that the characteristics of the rubber tree trunk, such as the tapping panel, front line, back line, hollow, and crack, can be recognized and have optimal coincidences.

In MATLAB, the repeated contour curves were matched using different colors, overlapped and exhibited in comparison to the differences in the contour curves after repeated acquisition. It can be noticed that the repeating contour curves may rotate or offset around the origin of the acquisition track, which could be related to data transmission delay and trunk surface roughness. However, as shown in Figure 8, this divergence has no effect on the contour curve results. In the meantime, changes in the relative position of the rubber tree trunk only affect the center point of the contour curve, not its accuracy.

The existence of hollows and cracks, as illustrated in Figure 8, will interfere with laser transmission, resulting in variances in acquisition results and even feature deformation or difficulty in identification, as indicated by the red arrow in Figure 8. This is most likely due

to a shift in the angle of incidence of the laser light as the laser finder passes through the location of the crack and hollow. After removing the feature distortion or undiscovered circumstances produced by data distortion, there are eight distortions in total, and the DCR is 94.67%.

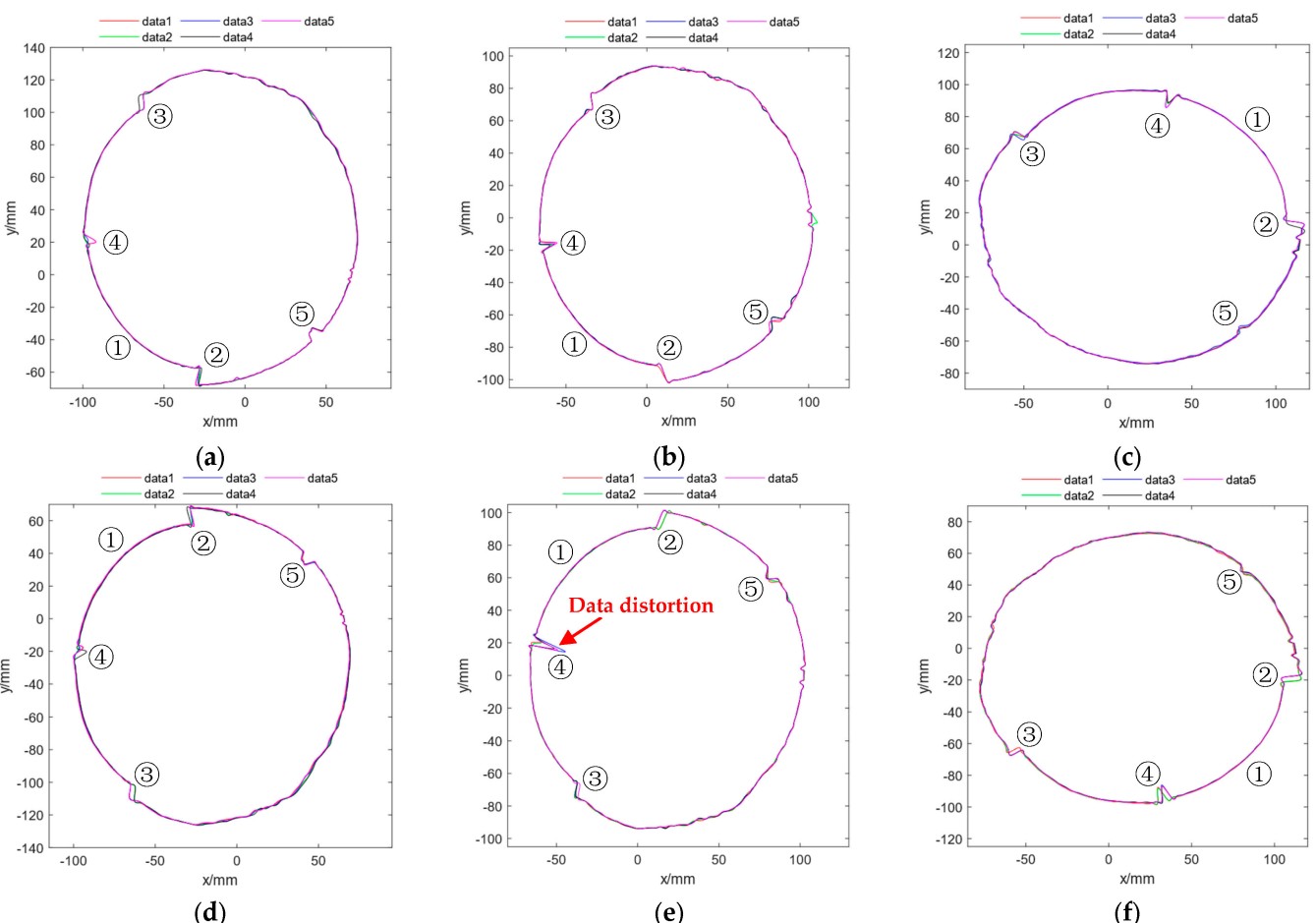

**Figure 8.** Results of different relative positions: (**a**,**d**) results of different directions in the first position; (**b**,**e**) results of different directions in the second position; (**c**,**f**) results of different directions in the third position.

The DRC results indicate that the characteristics of the rubber tree trunk could be recognized correctly and the repeatability of the experiments was optimal. The DRC is a critical factor in locating and planning the trajectories of automatic tapping machines. In general, the tapper can dynamically alter the location and recognition information in the manual tapping process, whereas this information is fixed in automatic tapping machines. Thus, the characteristics of starting points (front tapping line), finishing points (back tapping line), tapping line, and depth must be detected precisely. Incorrect starting points will cause the tapping machine to collide with the rubber tree, destroying the tapping device; incorrect finishing points will cause the latex to overflow, and the latex will be unable to be collected into a latex bowl along the back tapping line, allowing it to mix with large amounts of foreign matter and thus affecting its quality. Furthermore, leak recognition from cracks and hollows makes controlling tapping depth difficult, and an erroneous tapping depth not only reduces the production of natural rubber, but also harms the rubber tree, as shown in Figure 9.

Some characteristics cannot be expressed accurately and the point cloud data are distorted, which leads to missed leak detection. Hence, more research is needed to improve the detection accuracy, such as by condensing the error data and improving image features.

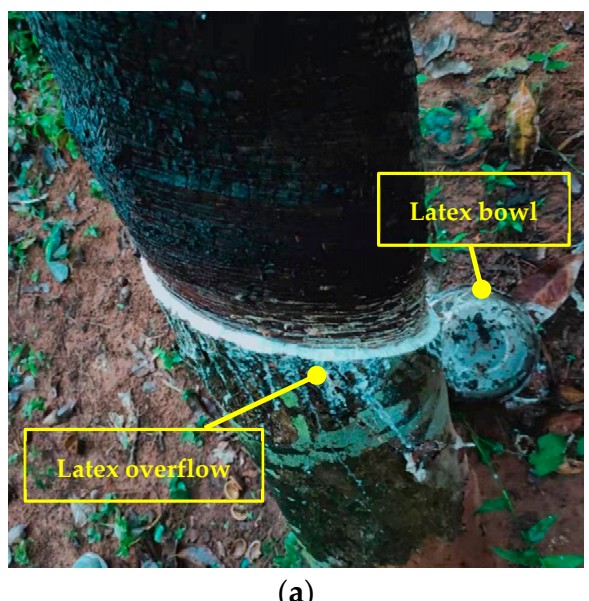
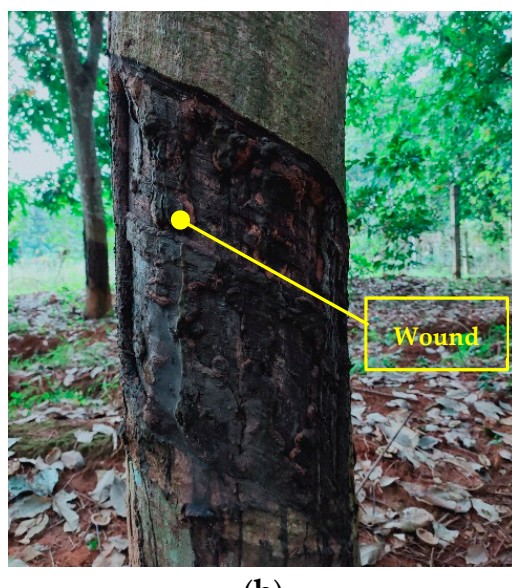

**Figure 9.** Phenomenon of incorrect tapping process: (**a**) phenomenon of latex overflow; (**b**) a wounded rubber tree.

### 3.3. Results of Statistical Analysis

In this study, the area of the trunk's contour curves was compared to assess the accuracy of the repeated extraction. The area of the closed curve can be determined in MATLAB thanks to the contour curves' closed attribute. The area data's variation coefficient was calculated and examined using means of mathematical statistics and variance analysis, as shown in Table 1. As indicated in Table 2, a two-factor analysis was conducted in SPSS to explore the impact of the relative position and acquisition direction.

**Table 1.** Results of variance analysis of contour curves' areas.

| Relative Position | Direction | Groups | Mean/$\mu$ | Standard Deviation/$s$ | Variable Coefficient/$\sigma$ |
|---|---|---|---|---|---|
| First position | Clockwise | 5 | 24,588.59 | 6.27 | 0.0003 |
| | Counterclockwise | 5 | 24,597.22 | 11.42 | 0.0005 |
| Second position | Clockwise | 5 | 24,689.61 | 8.71 | 0.0004 |
| | Counterclockwise | 5 | 24,688.65 | 18.33 | 0.0007 |
| Third position | Clockwise | 5 | 24,683.41 | 11.64 | 0.0005 |
| | Counterclockwise | 5 | 24,701.60 | 9.96 | 0.0004 |

**Table 2.** Results of two-factor analysis.

| Factor | Mean Square Error | $F$ | $p$ (<0.05) |
|---|---|---|---|
| Direction | 557.46 | 3.193 | 0.086 |
| Position | 57.02 | 0.253 | 0.621 |
| Direction × Position | / | 0.977 | 0.452 |

The repeated accuracy of the trunk's contour curves using the acquisition system proposed in this study is high and stable, as shown in Table 1, where the mean variability coefficient of the areas of the contour curves in different directions and positions is 0.04%. The good repeated accuracy indicates that the rubber tree trunk's contour curves that were collected before the tapping process are precise, allowing for the effective avoidance of the cumulative error of the continual tapping process.

In the clockwise direction, the mean standard deviation of the area of the trunk's contour curve gathered is 8.87 (lower than the overall mean), while in the counterclockwise direction, it is 13.24 (higher than the overall mean). An accurate and stable trunk contour curve can be obtained in the clockwise direction, even though there is little difference between the different directions. Additionally, as the acquisition experiments in different directions were conducted alternately, it follows that the data swing regularly with changes in acquisition direction from top to bottom. Although altering the collection direction has little impact, it may be preferable to guarantee consistency throughout the actual collection process.

It is also observed that the variable coefficient of the contour curves from different acquisition directions in the same relative position differs by 0.02%, proving that the impact of changing the acquisition direction is minimal. Consequently, it is unnecessary to adjust the position of the acquisition device before beginning the collection process, greatly enhancing the acquisition efficiency.

The confidence coefficients ($p$) of direction, position, and direction $\times$ position are 0.086, 0.621, and 0.452, respectively, based on the findings of the two-factor analysis. The results further indicate that the extraction is not greatly affected by changes in direction or positions, which implies that the contour curve can be extracted as long as the rubber tree is within the detection range of the laser range finder.

The continuous tapping process requires the accuracy of the repeated acquisition of contour curves. The tapping process is often performed every 2–3 days, and each operation should be based on the tapping line and tapping panel of the preceding operation, requiring less error from repeated acquisition and less impact from external factors. The aggregation of each contour curve's extraction error results in isolated rubber tree bark and an irregular tapping panel, as shown in Figure 10. Because the bark around the isolated bark is cut off, the nutrient channels in the tree are disrupted, and the isolated bark no longer produces latex, harming the growth of the rubber tree [40].

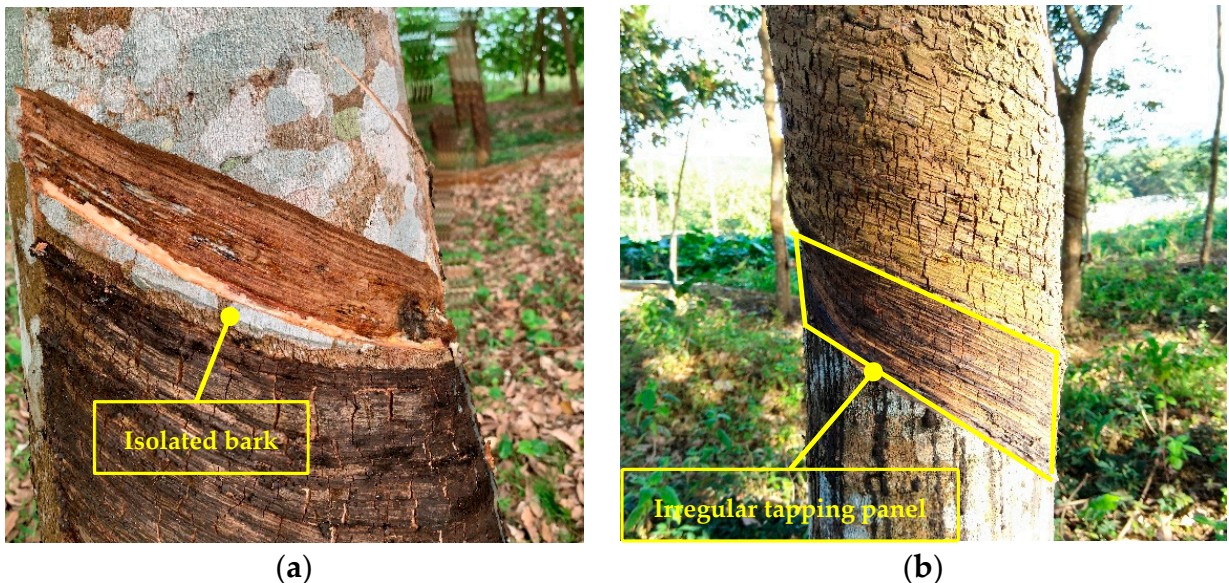

(**a**)  (**b**)

**Figure 10.** Effect of low precision of repeated tapping: (**a**) the isolated rubber tree bark; (**b**) the irregular tapping panel.

The results of repeated extraction also indicate that accuracy can be maintained during multiple acquisitions and that the effect of different positions and directions is insignificant, proving that the accuracy of repeated contour curve acquisition is high and fully meets the requirements for an automatic tapping machine. During the experiment, it was discovered that the imaging effects of deep cracks from multiple extracted contour curves can differ, resulting in distortions in the point cloud data (as shown by the red arrow in Figure 8).

The error completely throws off the crack depth estimation and the precision of the rubber tapping process. A preferable approach is to filter and screen the point cloud data from the crack's location before transmitting it to MATLAB to fit the contour curve, which will not have a significant impact on the extraction results. Simultaneously, we observed that the acquisition speed and data transmission frequency have an impact on the contour curve. Because of the radius of the laser point (as shown in Figure 11), slower speeds and faster frequencies will generate repeated collection points, whereas faster speeds and slower frequencies will reduce the fitting accuracy of the contour curve, resulting in a poor contour curve. Therefore, further research needs to be conducted in the future.

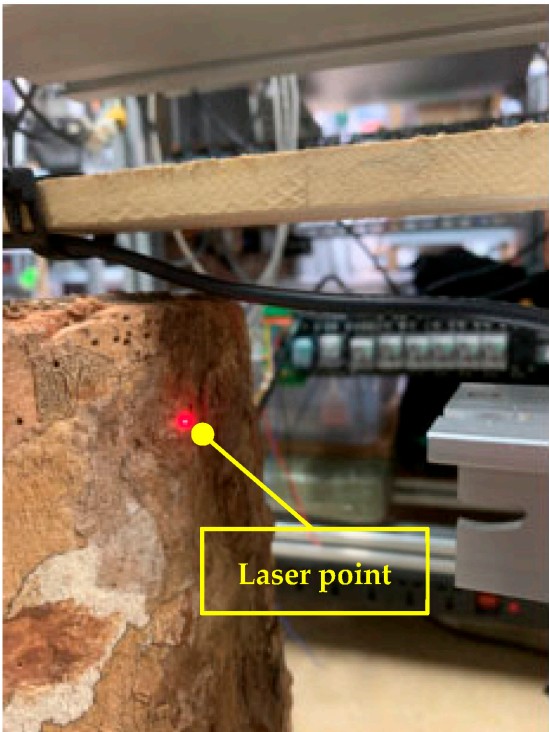

**Figure 11.** Radius of laser point during acquisition operation.

### 4. Discussion

*4.1. Acquisition System Performance*

The point cloud data of the rubber tree trunk were collected using the laser range finder. Its great accuracy and stability provided precise contour curve data. In this study, we established an acquisition system for such a laser range finder by extracting the contour curve of the rubber tree trunk.

*DCR* is a critical index of the acquisition system's identification accuracy. The *DCR* acquisition system proposed in this study achieves 94.67% accuracy and can identify the front tapping line, back tapping line, and tapping panel completely, ensuring the accuracy of the automatic tapping machine. It is critical to determine that the changes in direction and position have no relationship with the results because the relative position of the tapping machine and the rubber tree will alter as the rubber trees grow. If position changes have little influence on extraction, the tapping machine can retain its operating accuracy over time.

$\sigma$ is a measure of the disparity between the results of the contour curve area in the repeated experiment. In this study, the variation in $\sigma$ could be explained by the contour curve extraction's error. For instance, the highest $\sigma$ (0.0007) was found in the second relative position, while the lowest $\sigma$ (0.0003) was found in the first relative position, which might serve as a reference for the installation of the automatic tapping machine.

The contour curve of the rubber tree trunk is extracted using the B-spline fitting program. One advantage of this program is its simplicity, which can reduce the error of the tapping information. The average extraction time of the contour curve is less than 2 s (approximately 218 points). When this program is applied to an automatic tapping machine, only one embedded development board is required, and the application has no effect on the construction or weight of the automatic tapping machine. Overall, the acquisition system established in this study performs admirably when combined with practice production.

The acquisition plane of the rubber tree trunk's contour curve is parallel to the horizontal plane; however, the actual tapping plane (the acquisition plane) should be at an angle of around 22° from the horizontal plane. Moreover, the acquisition system was not tested in a rubber plantation, which has a complicated environment, and the surface of the rubber tree trunk was in worse condition than the experimental trunk. Therefore, it is necessary to carry out acquisition experiments in a rubber plantation in order to further improve the accuracy of the acquisition system.

### 4.2. Feasibility

The extraction of contour curves and surface features from rubber trees is essential for the development of automatic tapping machines. The contour curves can provide accurate location information and sensible tapping trajectories, guiding the machine to complete the tapping process along a specified route. An automatic acquisition system based on laser-ranging technology was built in this study, and the contour curve of a rubber tree trunk was extracted in MATLAB using the B-spline fitting approach, as shown in Figure 8. Although the extraction experiment was not carried out on a rubber plantation, it did provide a way for collecting rubber tree contour curves.

To further verify the feasibility of this acquisition system, a comparison of different extraction technologies, such as laser range, ultrasonic, and visual identification, is performed, as shown in Table 3.

**Table 3.** Comparison of different extraction technologies.

| Technology | Laser Range | Ultrasonic [41] | Visual Identification [42,43] |
|---|---|---|---|
| Measuring media | Laser range finder | Ultrasonic depth finder | Industrial camera |
| Prior knowledge | No need | No need | The knowledge of image processing |
| Objective | Information of contour | Information of contour and interior structure | Identifying species and contour |
| Position adjustment | No need | No need | Yes |
| Input data | Point cloud data | Scatter points | Pixels |
| Processing Method | Control point fitting | Control point fitting | Image filtering and processing |
| Output results | Curve, surface, or volume of contour | Surface or volume of contour | Image of contour |
| Efficiency | ≈2 min (Depends on quality of data) | ≈30 s (Depends on detection environment) | ≈2–4 s (Depends on quality of sample set) |
| Cost | <RMB 10,000 | >RMB 120,000 | >RMB 200,000 (Includes deep learning software) |

The low cost of this technology is a considerable advantage, especially given the state of the natural rubber industry's economics. The acquisition efficiency of this technology is rather low since sufficient data are required to extract the correct contour curve and the acquisition operation must maintain stable status and velocity, requiring additional time to finish the acquisition process. Although the visual identification technology reaches

maximum efficiency after collecting a large number of samples, it needs an adequate investment at the beginning, and is typically employed to identify species in the forest field. Only by lowering the economic cost can the rate of financial return and use of rubber tappers be improved. Using our model, there was no need to search for the center point (also known as datum), which is the coincidence point between the center of the trunk and the acquisition track during the entire experiment, greatly simplifying the operation of contour acquisition. Because the above techniques serve distinct purposes, the accuracy of the probe was not compared. Generally, the proposed method is more precise and economical when the accuracy of the contour curve is ensured.

*4.3. Applicability*

The advantages of adopting laser-ranging technology over other detecting technologies are point cloud data collection simplicity and acquisition system construction cost savings. The acquisition system utilized in this study collected contour curve information for one rubber tree in 2 minutes. As a result, at least 30 rubber trees (covering approximately 666 m$^2$) could be scanned in an hour. The contour curve information of the rubber trees may be collected in 2 minutes if each rubber tree is equipped with one acquisition system. Manual tapping, on the other hand, cannot provide precise contour curve information.

The application of the acquisition system to the automatic tapping machine is primarily determined by the structure of the acquisition system and the surface condition of the rubber tree trunk. Various kinds of automatic tapping machine are currently being deployed in rubber plantations to conduct tapping tests [10], although these machines are still not widely used. The reason for this is that these tapping machines are expensive and do not satisfy the standards. According to the acquisition system used in this study, the laser range finder can be loaded at the start of implementing the system. First, the cloud point data collected by the laser range finder will be processed using a fitting program before tapping; second, the tapping information of the rubber trees will be sent to the control system of the tapping machine; and finally, the implementing system of the tapping machine will execute the tapping process based on the tapping information, as shown in Figure 12. Although acquiring a contour curve takes time, this method can improve the operational precision of the tapping machine and eliminate the problem of a scarcity of rubber tappers.

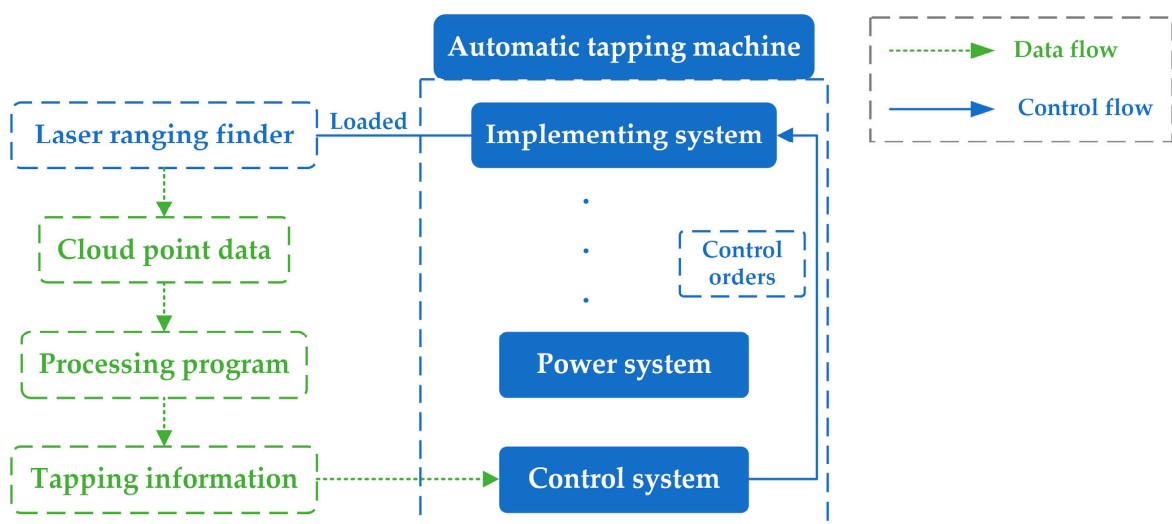

**Figure 12.** Application of the acquisition method.

Many impurities are attached to the surface of the rubber tree trunk due to the uncontrolled environment of the rubber plantation, affecting the accuracy of the acquisition system. Consequently, the primary objective for future studies is to focus on dealing mathe-

matically with the imperfect surface of the rubber tree trunk and filtering interference from the cloud point data.

## 5. Conclusions

The point cloud data of rubber tree trunks were acquired in the laboratory using laser-ranging technology, and the contour curve was extracted using the B-spline fitting program in MATLAB. The main conclusions of this study are as follows:

(1) The automatic acquisition system and B-spline fitting program proposed in this study have equivalent functions that enable the automatic contour curve extraction of rubber tree trunks. The contour curve of the rubber tree trunks can be used as a reference for the trajectory planning of rubber tapping equipment, and the B-spline fitting program is ideal for the extraction of irregular curves such as the rubber tree trunk's contour curve.

(2) Changes in acquisition directions and relative positions had no effect on the contour curves of the trunk, implying that the contour curve can be extracted as long as the rubber tree is within the range of the laser range finder. The acquisition system presented in this study is practicable, and its accuracy is high and reliable.

(3) The acquisition system in this study has the advantages of simplicity and convenience. It was unnecessary to adjust the position of the acquisition device prior to the collection process, which helped to improve acquisition efficiency. Because of its cheaper cost, this system can satisfy the development form of the rubber industry.

In general, the successful extraction of the rubber tree trunk's contour curves is important for the actual production and application of agroforestry. To begin, the acquisition system should be combined with the tapping machine and collect precise tapping information in order to modify the operating mode and tapping trajectory in real time and assure the precision of each tapping process. Second, the contour curve contains diameter and volume at breast height information, which is helpful to estimate the wood volume of rubber trees. Finally, this acquisition system strategy can be applied in various domains, such as the cultivation of longan and lychee fruit plants. These crops require the ring cutting at the stems to promote fruit bearing, and it is important to obtain the contour curves of the stems for the ring cutting machine.

**Author Contributions:** Conceptualization, T.L. and Y.Z.; methodology, T.L. and Y.Z.; software, T.L. and Y.Z.; validation, T.L. and Y.Z.; formal analysis, T.L. and G.W.; investigation, T.L. resources, T.L.; data curation, T.L.; writing—original draft preparation, T.L. and L.W.; writing—review and editing, T.L. and J.C.; visualization, C.H. and Y.Z.; supervision, Y.Z.; project administration, T.L. and Y.Z.; funding acquisition, T.L., G.W. and J.C. All authors have read and agreed to the published version of the manuscript.

**Funding:** This research was funded by the Hainan Provincial Natural Science Foundation of China, No. 323QN273 (T.L.), the Hainan Provincial Natural Science Foundation of China, No. 320QN349 (G.W.), and the Central Public-Interest Scientific Institution Basal Research Fund, No. 1630022022005 (J.C.).

**Institutional Review Board Statement:** Not applicable.

**Informed Consent Statement:** Not applicable.

**Data Availability Statement:** Not applicable.

**Acknowledgments:** All authors would like to express their utmost gratitude to the Xiamen Haiwell Technology Co., Ltd.

**Conflicts of Interest:** The authors declare no conflict of interest.

**Appendix A**

| Line | Program |
|------|---------|
| 1 | % 1-Converting cloud points to rectangular coordinate points |
| 2 | A = xlsread('record3.xlsx'); |
| 3 | Cloudpoint = A(:, 2); % Cloudpoint: Cloudpoint of rubber tree trunk |
| 4 | Rt = double(input("Please input the radius of rotate arm:")); |
| 5 | r0 = double(input("Please input the center distance:")); |
| 6 | r = Rt− (r0−Cloudpoint); % Polar diameter |
| 7 | N = length(Cloudpoint); % Volume of point cloud data |
| 8 | theta = (0:2*pi/(N−1):2*pi)'; % Polar angle |
| 9 | [x, y] = pol2cart(theta, r); |
| 10 | % 2-Using Thomas-method to calculate the control points |
| 11 | k = 3; % Degree of B-spline |
| 12 | A1 = eye(N)*4; |
| 13 | A1(1, N) = 1;A1(N, 1) = 1;A1(1, 2) = 1;A1(N, N−1) = 1; |
| 14 | for i = 2:N−1 |
| 15 | A1(i,i−1) = 1;A1(i,i + 1) = 1; |
| 16 | end |
| 17 | b1 = x*6; |
| 18 | cpx =Chase_method(A1,b1); % Abscissa of control points |
| 19 | b2 = y*6; |
| 20 | cpy =Chase_method(A1,b2); % Ordinate of control points |
| 21 | % 3-Fitting the curve of rubber tree trunk |
| 22 | n = 1; % Number of curve segments |
| 23 | for i = 1:N−3 |
| 24 | for u = 0:1/(N + k + 2):1 |
| 25 | BF0 = 1/6*(1−u)^3; |
| 26 | BF1 = 1/6*(3*u^3−6*u^2 + 4); |
| 27 | BF2 = 1/6*(−3*u^3 + 3*u^2 + 3*u + 1); |
| 28 | BF3 = 1/6*u^3; |
| 29 | % Fitting curve |
| 30 | x(n) = BF0*cpx(i,1) + BF1*cpx(i + 1,1) + BF2*cpx(i + 2,1) + BF3*cpx(i + 3,1); |
| 31 | y(n) = BF0*cpy(i,1) + BF1*cpy(i + 1,1) + BF2*cpy(i + 2,1) + BF3*cpy(i + 3,1); |
| 32 | n = n + 1; |
| 33 | end |
| 34 | end |
| 35 | x(end + 1,:) = x(1,:); |
| 36 | y(end + 1,:) = y(1,:); |
| 37 | plot(x,y,"r") |

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
