# Peer review of "Automatically Extracting Rubber Tree Stem Shape from Point Cloud Data Acquisition Using a B-Spline Fitting Program"

_forests, doi:10.3390/f14061122_

Round 1
Reviewer 1 Report
Dear Autors and Editor,
I had the pleasure of reading the article “ Automatically Extracting the Contour Curve of Rubber Tree based on Laser-ranging Technology”
Overall the article has the correct structure. The text touches on the matter of automatic extraction of rubber tree curves. I think the idea is quite interesting. The authors described this experiment logically and I didn’t have any problems understanding the idea. One of the important things for me is the cost of this equipment. In my opinion, such construction will be much cheaper than commerce LiDAR scanners. So in this area, authors can highlight additional positives of their experiments. I see some small errors in the text which can be improved but overall merit is good.
I wish authors further succeed in publication. Some small comments are presented below.
121
Move the description of the elements (1, 2, etc) to the figure 2 description.
128
“distance between contour point and center point O expressed in mm “
169
I’m not sure it is needed to put the program lines into the text. I would suggest moving into the appendix.
215
I don’t understand the idea of the STL structure program table. Is it really needed according to the discussion, conclusion, or results section?
224
Definitely, I don’t think table 3 I needed. You can write one sentence which covers this subject. It would be much clear.
225
Why there is only x and y moves and no rotation?
245
You shouldn’t start the chapter with the graph. By the way, the is no explanation of what colors mean and there are no units on the axis.
My general comment on the whole text touches on the word Anticlockwise – as far as I know, it should be replaced with counterclockwise.
267
Exactly the same comments as above are in Figure 9.
293
Do not start the chapter with a table. Why there are no units for the columns?
296-298
It is really strange to me that you used CAD for area calculation. There are better programs to do that (GIS). It also is connected with spline fitting. It is unclear if this spline was created automatically or manually.
311
In my opinion, the analysis is the weakest point of this text. I would suggest some improvements. Use graphs, try to compare them with some factors which for sure are. Do some statistical tests to confirm the significance of the results.
Author Response
Dear reviewer, Thanks for your comments which really useful to improved the research. Please allow me to introduce that the cost of this equipment is about 1500 RMB, and we thought the cost will increase to about 10000 RMB after upgrading the equipment. Finally, thank you again. The response of errors as shown in the submitted document. Best wishes.
Reviewer 2 Report
MS: Automatically Extracting the Contour Curve of Rubber Tree based on Laser-ranging Technology
This study develops a protocol for extracting the contour curve of rubber tree using Laser-ranging Technology. I believe Laser technology associate with Laser range finder technology instead of ground LiDAR? If authors have some photographs on taking measurement using Laser range finder may be useful to add them as a Figure. How quicker using Laser range finder to detect contour curve instead of observing through human eye. May be better justification is important to reader to understand the importance of the method.
Line 7: An introductory sentence is not complete i.e. strategic and raw materials used in ….. products,
Line 8 – 10: State Objectives clearly
Line 82: Yang et al. [29]
Line 77: Wu et al [27]
Line 74 : Ma et al. [26]
Line 86: Zhang et al. [30]
Line 88: Chen et al. [31]
Line 113: Label three Pictures as (a), (b) and (c) and give a brief Figure caption.
Line 245: Need a results narrative first prior to Figure 8, because Fig. 8 should cite in the main text with a narrative before presenting the figure.
Line 272-273: Give a caption on what red arrow stand for?
Line 267: Need a results narrative first prior to Fig. 9. So, move Line 274 – 285 just after the sub section heading.
Figs. 8 & 9: Label x and Y axis of the Figures
Line 308: extra space i.e. effection
Line 312: No literature cited in the Discussion. Make comparison among different automated contour curve methods.
Line 353 – 354: Can you elaborate how extraction of contour curve of rubber is impacted trajectory planning of rubber tapping equipment. This information is good for discussion.
Author Response
Dear reviewer, Thanks for your comments which were useful to improved the research, and I have added image of taking measurement. Please allow me answer to you. In general, the detection spped of tapper eyes is more quicker than the method proposed in this study, because the tapper can find the tapping points at a glance. The reason why we conducted this research was that we wanna find a probe method for automatic tapping machine, which is replacing the tapper because the lack of resource of tapper. Finally, thank you again. The response of some errors as shown in submitted document. Best wishes.
Reviewer 3 Report
Summary: Thank you for the opportunity to review the manuscript titled “Automatically Extracting the Contour Curve of Rubber Tree based on Laser-ranging Technology”. The paper used point cloud fitting technique to extract rubber tree trunks. The point cloud is fit using B-spiling fitting algorithm in Matlab software. The results show a 94.67% accuracy and is able to characterize the trunks regardless of relative positions, acquisition device and the direction of the measurements. The study suggests that this method is effective and efficient in locating specific structural characteristics of the tree trunk (hollows, cracks, tapping line etc.)
Broader comments: Thank you for the opportunity to review the manuscript titled “Automatically Extracting the Contour Curve of Rubber Tree based on Laser-ranging Technology”. The paper used point cloud fitting technique to extract rubber tree trunks. The point cloud is fit using B-spiling fitting algorithm in Matlab software. The results show a 94.67% accuracy and can characterize the trunks properties regardless of relative positions, acquisition device and the direction of the measurements. The study suggests that this method is effective, low cost, and can be incorporated with tapping instruments. The written language specially in the introduction need to be restructured as it mostly like a blog post rather than a manuscript. Further, in-text citation needs to be checked and formatted in a consistent way. Per title and the abstract, the manuscript should mainly be about the reconstruction of rubber tree trunk structural properties based on point cloud generated using range finder, which can be instrumented and automated later. However, only a very little information is provided in the method how the point cloud was collected, processed, what is the point density, how that is affected, and B spline parameters. Please add those parameters in the next version. Detailed information has been provided about the acquisition system, I believe is not the focus of the study as per title and the abstract. Further, study design and accuracy assessment methods have some caveats and does not show the effective of the method in terms of how good the method detecting tree truck structural properties in space. Rather the accuracy assessments described in this study show only some qualitative measures on how good identifying number of interested properties of the truck. More specific comments are as below and requires addressing them before publishing as the answers to those comments make the article more compelling and novel. The length of the paper could be reduced if focused only the specifics that are needed for the focus of the study.
Scientific soundness and written language: The manuscript more of like a blog and the sentences, their structure, and the wording need to be re-structured.
e.g. Line 67-70, 91-93. These are only a few specific examples.
L 166: What is Chase method (reference?)
L222- 223: Based on what parameters the position was changed? How much distance or angle from other positions? How were those changes decided and based on what assumptions?
L 233-234: Recognition of number of points will not equal to recognition of where those features are actually at in the tree trunk. I think, it is important to know correctly in space those features are recognized so an instrument can detect those feature locations in the future for process automation. I believe this study need to show those estimates to tell this method is effective.
Equation 1: How is the ∆L equals to the (r + ∆L)? (∆d= R – ∆L= R – (r + ∆L))
There is something wrong with the equation. Also, representation of r in the figure is not similar to the description of r in the text or at least not clear to the reader. Please clarify or correct.
Figures: Manuscript has less figures and information on how the laser point cloud or B-spline helped to improve detecting those structures rather the manuscript presents more on instrumentation. Can replace figures or remove figures 4,5, and 6.
Figure 8: Based on this there is a huge impact on tree position in fitting the B-spline in space. Why that is and need to be evaluated in the discussion as this could be huge problem if this issue is not resolved when plan to use this method in industry applications.
References: Check the reference order. In text citations do not match with the reference list. Also, please check how the in-text citation rules as per journal.
Author Response
Dear reviewer,
Thanks for your comments, which are very helpful for us to improved our manuscript and the next research. Please allow me to explain that the acquisition system and B-spline fitting program proposed in this study are mainly verify the feasibility and accuracy of this method so that the deeper studies have not been carried out in this study. Limited by the function of acquisition devices, there were still some shortages of this acquisition system. However, we are upgrading the acquisition system, the further research on the extraction of characteristics in the space will be conducted in the rubber plantation.
Thank you again, we will do better in the future research. Our response of the comments are shown in the document.

Reviewer 4 Report
The comments can be found in the file herewith attached.

Author Response
Dear reviewer,
Thanks for your comments, which are very useful for our manuscript. We have submitted the revision of the manuscript, and our response of the comments is shown in the document.

Reviewer 5 Report
This is a difficult manuscript to read, due to the many grammatical issues. I have noted some of the specific issues below, but the paper needs a strong edit. The references to papers are not handled according to protocol; for example line 48 the study by Hosoi et al. is cited, and, the text should say Hosoi et. al [14] transformed instead of Hosoi transformed, given there are multiple authors on the study. In addition, the number should follow the name directly.
The manuscript’s objective could be better defined, and the text needs to be objective throughout. Lines 69-70 “Considering of high accuracy of rubber tapping, it is better using laser-ranging scanning method to collect point cloud data of rubber tree.” sound like a hypothesis to be proven in this study, but it is instead probably an assumption, so that the authors then get to the objective in Lines 97-101. These sentences need to be written more clearly such as “In this research, we propose an automatic extracting method based on laser-ranging technology and conduct an experiment to test the feasibility of the method to support research of automatic rubber tapping equipment.” Or is the test simply of the feasibility of the method to support rubber tapping?” This is unclear. A strong edit is needed throughout to improve grammatical issues but also to focus on the proposed method.
Some grammar issues: Line 8: “…products, which is mainly…” (add the word ‘is’)
Line 6: “… , a repetitive extraction…” (add the connector ‘a’)
Line 20: “… the method is highly accurate and efficient.” Instead of … is high accuracy and efficiency.
Line 33: “The main reason” instead of “the mainly reason”
Line 40: “The critical element in developing accurate rubber tree contour curves is sufficient point cloud data.” Instead of the sentence that is there.
The entire introduction needs to be better summarized. Citing the publication: Yang, H.; Sun, Z.; Liu, J.;
Zhang, Z.; Zhang, X. The Development of Rubber Tapping Machines in Intelligent Agriculture: A Review. Appl. Sci. 2022, 12, 9304. https://doi.org/10.3390/app12189304 in the introduction would help get to the point quickly as to how this study fits in that framework.
The steps of the study look interesting and are generally explained well enough for a research journal. Streamlining the introduction, clarifying the objective, correcting the references within the text, and giving the manuscript a strong edit throughout is needed before it could be accepted for publication.
Author Response
Dear reviewer,
Thanks for your comments, which is very useful for us to improve the manuscript. Our response of the comments are shown in the document.
Thank you again.

Round 2
Reviewer 2 Report
Authors addressed most comments suggested and no further comments to ammend.
Author Response
Dear reviewer,
Thanks for your comments, we have improved the conclusion, as shown in the new manuscript.
Thank you again.
Reviewer 4 Report
This paper can be accepted in the present form.
Author Response
Dear reviwer,
It is glad to have your approval. Some English language and style have been moderated, as shown in the new manuscript.
Thank you again.
Reviewer 5 Report
The authors have addressed the comments. There is one place that needs clarification; other minor grammar issues are identified below and all need to be considered. Not all grammar issues may be listed, the manuscript could use another close read through to ensure grammar is correct.
The clarification which is needed is in lines 508-509: it is unclear what “...which is the key task for future research” is. Is the key task to figure out how to not have to clean the surface of the trunk? Please make clear what the key task is. If that is the key task, I would think it would make more sense to focus on dealing mathematically with the imperfect surface.
Suggested grammar improvements (not entire list):
Line 33: “..in the fields of aerospace,…”
L40: change diseases to diseased
L54: spell out what UAV stands for the first time it is used
L58: not sure what the word ‘almost’ means here. I think maybe replace almost with ‘many relevant’
L59: perhaps say tree stem location rather than tree detection.
L88: delete the phrase “to promote”
L112: change ‘is considered’ to ‘was adopted’
L143: instead of the word ‘described’ use ‘characterize’
L154: delete the word ‘finally’
L161: suggest deleting ‘firstly’
L218: suggest ‘perform’ rather than ‘realize’
L437: suggest using ‘simplicity’ instead of ‘simple’
L467: suggest ‘is in poor condition’ instead of ‘is in the doldrums’
L472: “…it needs adequate investment at the beginning” instead of ‘it needs to invest enough cost at the beginning…”
L473: instead of the ‘income and enthusiasm’ consider the ‘rate of financial return and use’
L492: missing citations—[XX]
L506: use “Due to the uncontrolled environment of the rubber plantation, there are many impurities attached to the surface of the rubber tree trunk which affects the accuracy...” instead of the sentence as written.
L511: “… of a rubber tree trunk based on laser-ranging technology in the laboratory…” instead of what is written
L524: use “.. simplicity and convenience.” Instead of simply and convenient.
L526: delete “a lot”
L55-536: “.. to promote fruit bearing…”
Author Response
Dear reviewer,
Thanks for your comments. Some English language and style have been moderated after using the closed read through and all the suggestions of grammar were accepted, as shown in the new manuscript.
Besides, we agree with your perspective on the key task. In Lines 504-505, we have clarified that the key task mainly focus on dealing mathematically with the imperfect surface and filter interference cloud point data in the future.
Thank you again.